# Evaluation of a Food Frequency Questionnaire for Capturing Free Sugars Intake in Australian Young Children: The InFANT FFQ

**DOI:** 10.3390/ijerph20021557

**Published:** 2023-01-14

**Authors:** Miaobing Zheng, Mihiri Silva, Stephanie Heitkonig, Gavin Abbott, Sarah A. McNaughton, Karen J. Campbell

**Affiliations:** 1Institute for Physical Activity and Nutrition (IPAN), School of Exercise and Nutrition Sciences, Deakin University, Geelong, VIC 3220, Australia; 2Murdoch Childrens Research Institute, Department of Paediatrics, The University of Melbourne, Royal Children’s Hospital, Parkville, VIC 3052, Australia; 3Melbourne Dental School, The University of Melbourne, Carlton, VIC 3053, Australia

**Keywords:** food frequency questionnaire, free sugars, early childhood, validation, dietary assessment

## Abstract

Excess free sugars intake contributes to dental caries and obesity in children. Food frequency questionnaires (FFQ) that assess free sugars intake in young children are limited. This study evaluated the utility of a 68-item FFQ to assess free sugars intake in Australian young children against three 24-h recalls at ages 1.5, 3.5, and 5.0 years. Free sugars intakes estimated from two methods were compared using group- and individual-level validation tests. Group-level tests revealed that mean free sugars intakes estimated from two methods were similar and Bland-Altman tests revealed no presence of proportional bias at age 1.5 years. For ages 3.5 and 5.0 years, the FFQ underestimated the free sugars intake compared to the recalls, and Bland-Altman tests revealed proportional bias. For individual-level tests, the deattenuated correlation (*R*) between free sugars intakes estimated from two methods exhibited good agreement across three time-points (R: 0.54–0.62), as were the percentage agreement (68.5–73.6%) and weighted kappa (K_w_: 0.26–0.39). The FFQ showed good validity at age 1.5 years. For ages 3.5 and 5.0 years, the FFQ showed good validity for individual-level tests only. The FFQ provided stronger validity in the ranking of individuals according to free sugars intake than comparing absolute free sugars intake at group level.

## 1. Introduction

Poor nutrition is a key factor contributing to the high prevalence of non-communicable diseases (NCDs), the leading cause of death and a major global cause of morbidity [1]. While poor nutrition may include a range of unhealthy dietary patterns, high consumption of free sugars has been identified as one of the potential risk factors for NCDs, including obesity, cardiovascular disease (CVD), type 2 diabetes, and insulin resistance [2,3,4].

Free sugars include ‘added sugars’, monosaccharides, and disaccharides that are added to foods and beverages, as well as sugars naturally present in honey, syrups, fruit juices, and fruit juice concentrates [5]. Unlike intrinsic sugars such as natural sugars in fruits, vegetables, and milk, which are likely to contain nutrients, fiber, and antioxidants, excess consumption of free sugars has been associated with a positive energy balance, leading to weight gain in the long term [3]. In addition, free sugars are the main cause of dental caries, the most globally common disease, affecting 2.4 billion people worldwide, at a cost of $642 billion [6]. Globally, intake of free sugars ranges from 7% up to 25%, with the highest intakes observed in children [7]. Considering the detrimental effect of free sugars in the development of obesity and dental caries, the World Health Organization (WHO) recommends that adults and children restrict free sugars intake to less than 10% of their total energy intake [5]. In addition, the WHO provides a further conditional recommendation to limit the intake of free sugars to below 5% of total energy intake, due to the progressive nature of dental caries and the likely lifelong impact of sugars [5]. While the majority of the burden of NCDs occurs later in life, compelling evidence suggests dietary patterns, including the intake of sugar-rich foods, are established early in life and may track into adult life, and have life-long impacts on health [8,9,10]. The impact of free sugars intake on dental health, food preference development, and obesity risk begins in childhood [5]. Measuring the intake of free sugars in children is, therefore, critical in developing effective early intervention programs and policies.

Findings from the 2011-2012 Australian National Nutrition and Physical Activity Survey revealed that high free sugars intake (estimated from 24-h recalls) is already evident in children aged 2–3 years old [11]. In Australian children, there is no food frequency questionnaire (FFQ) that assesses total dietary intake including free sugars intake. Only one validated FFQ (SMILE-FFQ) to date has been developed to specifically measure free sugars intake in children aged 18 to 30 months without an assessment of total dietary intake [12]. We previously documented the development and evaluation of a quantitative FFQ (InFANT FFQ) to assess overall dietary intake, including 26 food groups and 11 nutrients including total sugars but not free sugars [13]. In Australia, the Foods Standards Australian New Zealand has recently updated the Australian food composition database with the inclusion of free sugars content using a systematic protocol on the basis of analytical data and ingredients in food products [14], enabling additional validation of InFANT FFQ for assessing free sugars intake. Therefore, the present study aimed to evaluate the utility of the InFANT FFQ to assess free sugars intake in Australian young children against three 24-h recalls.

## 2. Materials and Methods

### 2.1. Study Design and Participants

This validation study was embedded in the Melbourne Infant Feeding Activity and Nutrition Trial (InFANT) program, which started as an early childhood obesity prevention trial [15,16,17]. At study baseline (2008), first-time parents and their 4-month-old infants were recruited from fourteen local government areas within the major metropolitan city of Melbourne, Australia. Details of the InFANT trial have been previously published [15,16,17]. In short, the trial consisted of a parent-delivered intervention focused on feeding and active play from the ages of 4 months to 1.5 years, with subsequent follow-ups (post intervention) at ages 3.5 and 5 years. Data from parent-child dyads who participated in the follow-ups at the ages 1.5, 3.5, and 5.0 years were used in the current study. At each follow-up, parents reported their child’s intake via a 68-item quantitative food frequency questionnaire (FFQ) and via three 24-h recalls for the purpose of this validation study [13]. Deakin University Ethics Committee (ID number: EC 175–2007) and the Victorian Office for Children (Ref: CDF/07/1138) granted ethical approval for the study. Parents provided written consent at each time point.

### 2.2. Free Sugars Intake from FFQ

The 68-item InFANT FFQ was developed using the 2007 Australian National Children’s Nutrition and Physical Activity Survey (NCNPAS), which was the most recent national dietary survey at the time of FFQ development. Details of the FFQ development process have been reported and a copy of the InFANT FFQ has been previously published [13]. In brief, food items and the median portion sizes of each food item among children aged 2–5-years-old were obtained from the 2007 NCNPAS, with food items selected based on contribution to key nutrients for obesity prevention and indictors of diet quality (energy, fat, saturated fat, total sugars, protein, fiber, folate, vitamin C, iron, and calcium). Foods items that explained the most variation (targeting 80% of variation) in intakes of key nutrients and identified by stepwise regression techniques were included in the FFQ [18,19]. The FFQ consists of an initial section with questions pertaining to general eating habits (e.g., type of bread, milk, supplement use) followed by a section asking about the frequency of consumption over the past month for 68 food items with 9 frequency response options (‘never or less than once a month’ to ‘6 or more times a day’). A purpose-designed database containing the nutrient profile for each FFQ item was developed by matching each FFQ item with one or more foods from the AUSNUT2007 food composition database. This was used to calculate food and nutrient intakes from the FFQ. Free sugars values were not included in AUSNUT 2007, and were therefore extracted from the subsequent food composition database (AUSNUT 2011–2012). An AUNUST 2007 and 2011-2013 matching file was used to ensure the optimal matching of food items [20]. The matching file was developed using food identification codes, existing matching files, as well as conceptual and nutritional similarities, and aimed to facilitate conversion of dietary data from AUSNUT 2007 to 2011–2013. Daily nutrient intakes including total energy intake and free sugars in grams for each participant were derived from daily equivalents converted from frequency of consumption over the past month along with median portion sizes.

### 2.3. Free Sugars Intake from 24-h Recalls

Within seven days after FFQ completion, parents also reported their child’s intake via three interviewer-administered 24-h recalls over three non-consecutive days, including two weekdays and one weekend day. The interview was conducted over the phone with trained dieticians using the five-pass United States Department of Agriculture standardized recall process [21]. Main carers reported all food and beverages consumed by the child in the previous day (past 24 h) and used a purpose-designed food model booklet to estimate portion size. Over 2000 food or beverage items were reported across three time points. Nutrient profiles for these items were obtained from matching similar foods (one or more) from the Australian Food Supplement and Nutrient Database (“AUSNUT2007”). Free sugars values for each item were adapted from AUSNUT 2011-2013. A dietician checked all recalls for accuracy and completeness. At each time point, the mean free sugars intake of the three 24-h recalls was compared to the free sugars intake calculated from the FFQ.

### 2.4. Child and Maternal Variables

Several variables were used to describe the sample: child age (years), sex (boys vs. girls), maternal education (low: completed up to year 12; intermediate: completed trade/certificate post-secondary school; high: completed university degree or beyond), and body mass index (BMI) z-scores. These covariates were collected at each follow-up. Child age, sex, and maternal education were collected via parental questionnaires. Trained staff measured children’s weight to 10 g using calibrated infant digital scales (Tanita 1582, Tokyo, Japan), and height/length to 0.1 cm using a calibrated measuring mat (Seca 210, Seca Deutschland, Hamburg, Germany) or a portable stadiometer (Invicta IPO955, Oadby, Leicester, UK). Age- and sex-specific BMI z-scores were calculated using WHO growth standards.

### 2.5. Analytical Sample

Consistent with our previous validation study [13], participants were included in this analysis if they had three 24-h recalls, FFQ data with <10% of missing items, complete data for covariates used to describe the sample (child age, sex, and BMI z-score and maternal education), energy intakes within ±3 SD, and had not consumed breastmilk. This resulted in final sample sizes of 231 (1.5 years), 172 (3.5 years), and 187 (5 years), respectively. A detailed flow chart showing detailed exclusions and the final sample sizes across three time points has been reported in the previous main FFQ validation study [13]. Comparisons of child sex, maternal education, and BMI z-score between the children included and those excluded from the analysis has been previously reported, with no significant difference being found. Sample characteristics have been previously reported [13]. Across three time points, there were close to equal proportions of boys and girls. The sample consistently comprised a higher proportion of mothers with tertiary education (55–62%). Mean BMI z-scores of children ranged from 0.6 to 0.8 units [13].

### 2.6. Statistical Analysis

Based on the recommendation by Lombard et al. [22], mean free sugars intakes estimated from FFQ and 24-h recalls were compared by several group-level and individual-level validation tests to assess different facets of validity. Group-level tests provide insights into the validity of the FFQ for group-level comparisons of absolute free sugars intake. Individual-level tests infer validity of the FFQ for the ranking of participants according to free sugars intake. Free sugars intakes from both FFQ and 24-h recall were energy-adjusted to remove correlated measurement error using the residual method [19]. Energy-adjusted free sugars intakes were used in all subsequent validity testing.

Several tests were performed to assess validity at the group level. Paired *t*-tests compared the mean free sugars intakes from the two methods, reflecting agreement at the group level. Bland-Altman correlation between the mean and mean difference of free sugars intakes estimated from FFQ and 24-h recall was undertaken to examine the presence of proportional bias and agreement at the group level (*p*-value ≤ 0.05 indicates presence of bias). The Bland-Altman limit of agreement (LOA) index represents the percentage of participants with values outside the LOA, which was calculated as the mean difference in free sugars intake ±1.96 SD. An LOA index < 5% is considered acceptable [22,23]. Bland-Altman plots were used to identify trends in bias and outliers. Linear regression between the mean and mean difference of free sugars intake from the two methods was conducted, with regression coefficients not equivalent to zero indicating a presence of proportional bias.

For assessing individual-level agreement, Pearson correlation (*r*) was performed to examine the strength and direction of association between the two measures at the individual level. To account for day-to-day variation in three 24-h recalls, the de-attenuated correlation coefficient was also assessed using the formula: $R=r\sqrt{1+\frac{{ʎ}_{x}}{n_{x}}}$, in which the attenuation factor (${ʎ}_{x}$) is the ratio of within- and between-person variances obtained from the three recalls and $n_{x}$ equal to 3 (three days of recalls). Correlation coefficients of <0.2, 0.20–0.49, and ≥0.5 were defined as poor, acceptable, and good agreement [21], respectively. Categorical agreement evaluates the cross-classification and ranking of participants, including chance, representing agreement at the individual level. Participants were grouped into quintiles and the proportion of participants correctly classified within one quintile was assessed, with ≥50% being considered as a good agreement [21]. Weighted kappa statistics were calculated to examine categorical agreement at the individual level, above that expected by chance, with the following criteria: poor (K_w_ < 0.20), acceptable (K_w_ = 0.20–0.59), and good agreement (K_w_ ≥ 0.60) [21,23]. All statistical analyses were conducted in STATA 17 (StataCorp, College Station, TX, USA) with significance level set at *p* < 0.05.

## 3. Results

Energy-adjusted mean and median free sugars intakes (g/day) estimated by FFQ and three 24-hrecalls at ages 1.5, 3.5, and 5.0 years are illustrated in Table 1. Results of group-level validation tests comparing mean intakes estimated by FFQ and 24-h recalls are presented in Table 2. At age 1.5 years, paired *t*-test showed FFQ and 24-h recall estimates of mean free sugars intakes were similar (*p* = 0.93), and the Bland-Altman correlation of mean and mean difference revealed no presence of proportional bias (*p* = 0.56). For ages 3.5 and 5.0 years, relative to 24-h recall estimates, FFQ underestimated the free sugars intake by approximately 20%, and Bland-Altman correlation revealed evidence of proportional bias (*p* < 0.001). At all three ages, there was an acceptable agreement for the Bland-Altman LOA index, with <5% of individuals having free sugars intakes outside of the limit of agreement. The Bland-Altman plots (Figure 1) and linear regression of the mean and mean difference of FFQ and 24-h recall also showed no evidence of bias at age 1.5 years (β = 0.01, *p* = 0.93), but there was presence of bias at ages 3.5 (β = −0.53, *p* < 0.01) and 5.0 years (β = −0.34, *p* < 0.01). The negative slope of bias indicates FFQ was more likely to underestimate free sugars intake against the 24-h recalls among individuals with higher free sugars intakes.

With respect to individual-level tests (Table 2), the deattenuated correlation (*R*) between FFQ and 24-h estimates exhibited good agreement across three time points, with correlation coefficients ranging from 0.54 to 0.62. Similarly, the percentage agreement of free sugars intake (within first quintile) showed a good outcome, with the highest agreement at age 1.5 years (73.6%), followed by age 3.5 years (71.5%) and 5.0 years (68.5%). Weighted kappa revealed acceptable agreement at all three time points (K_w_ 0.26 to 0.39).

## 4. Discussion

Describing young children’s free sugars intake is important given the detrimental influence of free sugars on child dental caries and health. Our study evaluated the validity of InFANT FFQ, an existing quantitative FFQ, for estimating free sugars intake relative to three 24-h recalls over three time points in early childhood. We used a combination of statistical tests to obtain comprehensive insights into different aspects of group and individual level validity. The InFANT FFQ showed good validity for both group and individual level tests at age 1.5 years. For ages 2.5 and 5.0 years, the FFQ showed good validity for individual-level tests, but not for group-level tests.

The good individual level validity at ranking participants across three time points supports the utility of the InFANT FFQ in assessing diet and disease relationships in epidemiological studies [24]. At the group-level comparing absolute intakes, FFQ and 24-h recalls estimated similar free sugars intake at age 1.5 years. However, the InFANT FFQ underestimated free sugars intake against 24-h recalls at ages 2.5 and 5.0 years. Moreover, the underestimation was higher in high free sugars consumers. The decline in group-level validity with age may be attributable to the increasing variety and intakes of free-sugars-containing foods with age. Given the InFANT FFQ was not intended to specifically measure free sugars intake, not all free sugars containing foods were captured. In contrast, given the open-ended nature of 24-h recalls, they may be more capable of capturing free-sugars-containing food sources. Moreover, FFQs in general tend to group food items into aggregated food groups, which may impact the estimation of some nutrients. For example, sugars, jam, honey, and syrups were grouped as one food item in the InFANT FFQ. Similarly, the InFANT FFQ has one item including all cakes, muffins, scones, and muesli bars. These foods were, however, reported as separate food items in the 24-h recalls. This is particularly relevant when free sugars content varies across aggregated food items, or if specific food items dominate consumption within food groups, as this would influence the ability of FFQ to estimate intakes when compared to 24-h recalls. It is also expected that intakes of free-sugars-containing food items would be low in 1.5-year-old children and increase with age. As such, the contribution of these food items to free sugars intake would also increase and lead to greater underestimation with age. As expected, our study revealed greater underestimation of absolute free sugars intake in the FFQ when compared to 24-h recalls in older ages at 2.5 and 5.0 years and in higher free sugars consumers.

FFQs that allow for the assessment of free sugars intake are limited. To the best of our knowledge, the InFANT FFQ [13] and the SMILE-FFQ [12] are the only FFQs enabling the assessment of free sugars intake in Australian preschool children. In contrast to InFANT FFQ, the SMILE-FFQ was developed to specifically measure total and free sugars intake in Australian toddlers aged 18–30 months, but not total diet. The SMILE-FFQ was developed based on dietary contributors and protectors to dental caries, incorporating a total of 89 food items including food and beverage sources of total sugars, free sugars, and fluoride, as well as protective foods including cheese and other milk products, chewing gum, and xylitol [12]. The InFANT FFQ, however, was developed based on national nutrition survey data and assesses both food groups and nutrients with a focus on including food items contributing to key nutrients relevant for obesity prevention and indictors of diet quality. Regardless of the differences in tool development, SMILE-FFQ also revealed stronger validity for capturing free sugars intake at the individual level than at the group level when validated against three 24-h recalls, as observed in our study.

Internationally, there are two other existing tools that have been used to assess free sugars intake. Mumena et al. [25] and Hunsberger et al. [26] developed and validated a FFQ to assess free sugars intake in 424 Saudi children aged 6–12 years and 99 Polish children aged 10–17 years, respectively. Consistent with findings from the present analysis of the InFANT FFQ, the 41-item FFQ developed by Hunsberger et al. revealed that free sugars intakes estimated from the FFQ and 24-h recalls were highly correlated, but the FFQ underestimated free sugars intake [26]. In contrast, Mumena et al. [25] found that their FFQ overestimated free sugars intakes relative to the 24-h recalls, and there was a low correlation between free sugars intakes estimated from the two methods. The overestimation in Mumena et al. [25] was likely due to the inclusion of a comprehensive list of free-sugars-containing food sources (12 food groups and 41 food items) as the FFQ was purpose-designed to capture free sugars intake. It is worth noting that the overestimation of dietary intakes is a well-known phenomenon in FFQ development [18]. The longer the list of foods, the more likely an FFQ will overestimate against 24-h recalls. For instance, this is commonly seen with estimations of fruit and vegetable intakes [27,28].

This study has several strengths and limitations. The InFANT FFQ is the first FFQ that enables the assessment of free sugars intake, along with other dietary food groups and nutrients in Australian preschool children. Adapting an integrative interpretation approach with a range of group and individual level tests enables in-depth exploration of varying facets of FFQ validity. These insights will inform decision-making on the selection of dietary tools for specific dietary-related questions and, in turn, interpretation and discussion of the results. Other strengths include the development of FFQ using national health survey data, a relatively large sample size, the use of three 24-h recalls, including both weekdays and weekend days as the reference method, and repeated validity testing at multiple ages in early childhood. Despite these strengths, there were a number of limitations that need to be acknowledged. In order to ease subject burden, the InFANT FFQ uses median portion sizes from the Australian national nutrition survey. This may have resulted in an underestimation of the free sugars in FFQ, as portion sizes of individual foods in 24-h recalls were reported by parents. One limitation of the current study is the lack of free sugars content from the AUSNUT 2007 composition database. The AUSNUT 2011-2013 food composition database was therefore adapted with the use of a AUSNUT 2007 and a 2011-2013 matching file to ensure accuracy [19]. Changes in food supply and free sugars content are possible and need to be acknowledged. It would be desirable to apply a scaling factor accounting for changes in free sugars content in the food supply from 2007 to 2011-2013 if such information is available. However, this would not influence the comparison of free sugars intake estimated from the two methods, as both use the same free sugars composition database. Notably, both 24-h recalls and FFQ are parent-reported, and reporting bias cannot be dismissed. Emerging research identified two promising biomarkers of dietary sugars intake: 24-h urinary sugar secretion and carbon isotope ratios (^13^C/^12^C ratio expressed as δ^13^C) [29,30]. Further FFQ validation studies could include the use of these biomarkers. Other limitations pertaining to the original FFQ development were previously reported in detail. For instance, the time gap between the two methods may result in a variation in intakes and, in turn, affect the comparability between the two methods. Moreover, the InFANT sample consisted of a higher proportion of mothers with tertiary education. This may influence the generalizability of InFANT FFQ to the wider Australian population.

## 5. Conclusions

This study tested the utility of the InFANT FFQ in capturing free sugars intake relative to three 24-h recalls. In summary, the InFANT FFQ provided stronger validity in ranking individuals according to free sugars intake than comparing absolute free sugars intake at the group level. The InFANT FFQ is a valid tool for assessing free sugars intake in Australian preschool children in epidemiological studies to explore associations between free sugars intakes and health outcomes. Future studies evaluating the validity of the InFANT FFQ for capturing free sugars intake in diverse populations with inclusion of biomarkers will be valuable.

## Figures and Tables

**Figure 1 ijerph-20-01557-f001:**
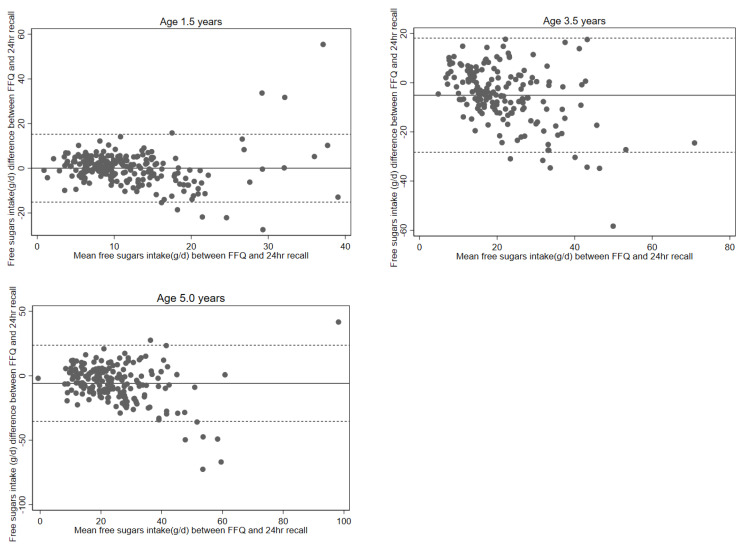
Bland Altman Plots comparing free sugars intake estimated by food frequency questionnaire (FFQ) and three 24-h recalls at ages 1.5, 3.5, and 5.0 years. Dotted lines as +/−2 SD.

**Table 1 ijerph-20-01557-t001:** Energy-adjusted free sugars intake (g/d) estimated by food frequency questionnaire (FFQ) and three 24-h recalls at ages 1.5, 3.5, and 5.0 years ^1^.

	Age 1.5 Years (n = 231)		Age 3.5 Years (n = 172)		Age 5.0 Years (n = 187)	
	Mean (SD)	Median (25th, 75th)	Mean (SD)	Median (25th, 75th)	Mean (SD)	Median (25th, 75th)
FFQ	11.7 (7.8)	10.2 (7.6, 14.2)	19.3 (9.2)	17.2 (13.8, 24.6)	21.9 (12.5)	19.8 (14.5, 26.8)
24-h recall 1	11.0 (8.1)	9.4 (5.3, 15.5)	24.6 (16.2)	21.1 (12.0, 34.8)	27.1 (23.6)	22.9 (12.3, 33.6)
24-h recall 2	13.0 (10.9)	10.2 (4.5, 19.3)	24.9 (20.9)	20.9 (10.8, 32.7)	28.1 (18.9)	24.0 (14.1, 38.5)
24-h recall 3	11.3 (10.3)	8.5 (4.9, 15.0)	23.9 (18.7)	19.5 (11.8, 32.3)	28.5 (24.4)	24.2 (12.1, 36.6)
24-h recall mean intake	11.7 (7.7)	9.6 (6.8, 15.2)	24.4 (14.0)	21.9 (14.4, 30.4)	27.9 (16.1)	25.0 (17.5, 35.4)

^1^ SD: standard deviation.

**Table 2 ijerph-20-01557-t002:** Group-level and individual-level comparison of energy-adjusted free sugars intake (g/d) estimated by food frequency questionnaire (FFQ) and three 24-h recalls at ages 1.5, 3.5, and 5.0 years ^1^.

	FFQ	24-h Recall	Group-Level Test			Individual-Level Test			
	Mean (SD)	Mean (SD)	Paired *t*-Test	Bland- Altman Pearson *r* *p*-Value	Bland- Altman’s LOA Index	Pearson Correlation *r*	Deattenuated Pearson Correlation *R*	Percentage Agreement	K_w_
Age 1.5 years	11.7 (7.8)	11.7 (7.7)	*p* = 0.93	*p* = 0.56	3.9%	0.50	0.56	73.6%	0.39
Age 3.5 years	19.3 (9.2)	24.4 (14.0)	*p* < 0.001	*p* < 0.001	4.1%	0.55	0.62	71.5%	0.33
Age 5.0 years	21.9 (12.5)	27.9 (16.1)	*p* < 0.001	*p* < 0.001	4.3%	0.47	0.54	68.5%	0.26

^1^ SD: standard deviation; LOA: limits of agreement; K_w_: Weighed Kappa.

## Data Availability

Data could not be shared publicly due to ethical reasons. However, data access is available from the corresponding author upon reasonable request.

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
