# Peer review of "Evaluation of a Food Frequency Questionnaire for Capturing Free Sugars Intake in Australian Young Children: The InFANT FFQ"

_ijerph, 2023, doi:10.3390/ijerph20021557_

Round 1

Reviewer 1 Report

Line 68: Is the sample of interviewees whose data were used in the validation of the questionnaire (described in the previous study) representative? If not, then this limitation should be highlighted in the discussion.

Lines 94-98: I suggest to write the link to the questionnaire instead of describing it

Lines 143-146: Results are not mentioned in materials and methods section. Here I believe that, if I understood correctly, the method from the previous study should be described or just referred to

218-220: Only the results should be discussed in the discussion section. Therefore, this sentence should be placed in methods.

250-253: These two sentences should be in the introduction section explaining why the InFANT FFQ and not the SMILE-FFQ was chosen for this analysis.

Line 260-274: The entire paragraph should be in the introduction section unless the results from the studies mentioned in this paragraph are compared (or can be compared) with the results from this study.

Author Response

Line 68: Is the sample of interviewees whose data were used in the validation of the questionnaire (described in the previous study) representative? If not, then this limitation should be highlighted in the discussion.

Response: Thank you for your time in reviewing our paper and we appreciate your comments and suggestions. Discussion about the representative of the cohort is described in Line 318-321: “Moreover, the InFANT sample consisted of a higher proportion of mothers with tertiary education. This may influence the generalisability of InFANT FFQ to wider Australian population.”

Lines 94-98: I suggest to write the link to the questionnaire instead of describing it

Response: A copy of the InFANT FFQ has been published in the main FFQ validation study as a supplementary file. We have now made this clear in line 94: “Details of the FFQ development process has been reported and a copy of the InFANT FFQ has been published previously [13].” We did not provide a direct link as it may result in copyright issues. Readers can access the FFQ from the open access original FFQ validation paper.

Lines 143-146: Results are not mentioned in materials and methods section. Here I believe that, if I understood correctly, the method from the previous study should be described or just referred to

Response: Sample characteristics are previously reported in the main FFQ validation paper. To avoid duplication of results publication, we included sample characteristics in the method (Analytical sample) to provide context for readers. We have now provided reference [13] to support sentences describing sample characteristics. Line 147-153

218-220: Only the results should be discussed in the discussion section. Therefore, this sentence should be placed in methods.

Response: Thanks again for the comment. We however respectfully disagree and believe the sentence “We have used a combination of statistical tests to obtain comprehensive insights into different aspects of group and individual level validity“ could be placed in the first paragraph of the discussion section as it highlights the methodological strength of the present validation study. The first paragraph of the discussion should not only provide an overall summary of the findings but also summarize the novelty, strength, and key implication of the paper.

250-253: These two sentences should be in the introduction section explaining why the InFANT FFQ and not the SMILE-FFQ was chosen for this analysis.

Response: The current analysis documents the validation of the INFANT FFQ for measuring free sugar intake. Our INFANT FFQ is the second FFQ that captures free sugar intake in Australian preschool children following the SMILE-FFQ. Therefore, a detailed comparison of the two FFQs is provided in the discussion. We modified the introduction in line 67-73 to explain why INFANT FFQ is used for present analysis: “In Australia, the Foods Standards Australian New Zealand has recently updated the Australian food composition database with inclusion of free sugar content using a systematic protocol on basis of analytical data and ingredients in food products [10], enabling additional validation of InFANT FFQ for assessing free sugars intake. Therefore, this present study aimed to evaluate the utility of the same FFQ to assess free sugar intake in Australian young children against three 24-hour recalls.” Also note that SMILE-FFQ cannot be used for the current analysis.

Line 260-274: The entire paragraph should be in the introduction section unless the results from the studies mentioned in this paragraph are compared (or can be compared) with the results from this study.

Response: We believe this paragraph is well-placed in the discussion section, as it did provide detailed comparison of the InFANT FFQ with the existing FFQs internationally that measures free sugars intake. The paragraph states that Hunsberger et al showed consistent findings with our InFANT FFQ. In contrast, Mumena et al showed contrasting findings and we provided potential reasons for the discrepant findings.

Reviewer 2 Report

The manuscript is written very well, and theme is very interesting; however, I recommend few minor changes:

1. The authors should firstly highlight the originality of the work, given that several studies with identical objectives have been carried out throughout the years.

2. The manuscript's overall presentation should be examined, as should the journal's criteria for the number of spaces between headings, subtitles, figures, tables, and text;

3. I recommend a revision of the manuscript according to the requirements of the journal (English, technical editing, bibliography, etc.);

Author Response

The manuscript is written very well, and theme is very interesting; however, I recommend few minor changes:

Thank you for your time in reviewing our paper and we appreciate your comments and suggestions.

  1. The authors should firstly highlight the originality of the work, given that several studies with identical objectives have been carried out throughout the years.

Responses: The originality of the work is highlighted in introduction line 58-73: Findings from the 2011-12 Australian National Nutrition and Physical Activity Survey revealed that high free sugars intake (estimated from 24-hour recalls) is already evident in children aged 2-3 years olds [11]. In Australian children, there is no food frequency questionnaire (FFQ) that assesses total dietary intake including free sugar intake. Only one validated FFQ (SMILE-FFQ) to date has been developed to specifically measure free sugar intake in children aged 18 to 30 months without assessment of total dietary intake [12]. We previously documented the development and evaluation of a quantitative FFQ (InFANT FFQ) to assess overall dietary intake including 26 food groups and 11 nutrients including total sugars but not free sugars [13]. In Australia, the Foods Standards Australian New Zealand has recently updated the Australian food composition database with inclusion of free sugars content using a systematic protocol on basis of analytical data and ingredients in food products [10], enabling additional validation of InFANT FFQ for assessing free sugars intake. Therefore, this present study aimed to evaluate the utility of the same InFANT FFQ to assess free sugars intake in Australian young children against three 24-hour recalls.

And discussion in line 290-291: “The InFANT FFQ is the first FFQ that enables the assessment of free sugar intake along with other dietary food groups and nutrients in Australian preschool children.”

  1. The manuscript's overall presentation should be examined, as should the journal's criteria for the number of spaces between headings, subtitles, figures, tables, and text;

Response: We have now checked the overall presentation of the manuscript, ensuring alignment with journal requirement. Happy to revise should there be any further changes required.

  1. I recommend a revision of the manuscript according to the requirements of the journal (English, technical editing, bibliography, etc.);

Response: We have now checked the overall presentation of the manuscript, ensuring alignment with journal requirement. Happy to revise should there be any further changes required.